# Cell Culture Systems and Drug Targets for Hepatitis A Virus Infection

**DOI:** 10.3390/v12050533

**Published:** 2020-05-12

**Authors:** Tatsuo Kanda, Reina Sasaki, Ryota Masuzaki, Naoki Matsumoto, Masahiro Ogawa, Mitsuhiko Moriyama

**Affiliations:** Division of Gastroenterology and Hepatology, Department of Medicine, Nihon University School of Medicine, 30-1 Oyaguchi-kamicho, Itabashi-ku, Tokyo 173-8610, Japan; sasaki.reina@nihon-u.ac.jp (R.S.); masuzaki.ryota@nihon-u.ac.jp (R.M.); matsumoto.naoki@nihon-u.ac.jp (N.M.); ogawa.masahiro@nihon-u.ac.jp (M.O.); moriyama.mitsuhiko@nihon-u.ac.jp (M.M.)

**Keywords:** Anti-HAV, drug screening, HAV, Huh7, IRES, PLC/PRF/5, subgenomic replicon, 5′ UTR, vaccine

## Abstract

Hepatitis A virus (HAV) infection is one of the major causes of acute hepatitis, and this infection occasionally causes acute liver failure. HAV infection is associated with HAV-contaminated food and water as well as sexual transmission among men who have sex with men. Although an HAV vaccine has been developed, outbreaks of hepatitis A and life-threatening severe HAV infections are still observed worldwide. Therefore, an improved HAV vaccine and anti-HAV drugs for severe hepatitis A should be developed. Here, we reviewed cell culture systems for HAV infection, and other issues. This review may help with improving the HAV vaccine and developing anti-HAV drugs.

## 1. Introduction

Hepatitis A virus (HAV) infection is a health problem in developing and developed countries [1,2]. HAV is transmitted mainly through the fecal–oral route or close physical contact [3]. HAV infection is associated with HAV-contaminated food and water [4,5]. An increase in HAV infection has also occurred among men who have sex with men (MSM) [6,7]. HAV infection occasionally causes acute liver failure and leads to death in HAV-infected patients [8,9].

HAV is a single-stranded positive-stranded RNA virus that circulates in blood as a membrane-cloaked quasi-enveloped virus (eHAV), but is shed in the feces as a naked, non-enveloped virion during acute hepatitis A virus infection [10]. The HAV genome consists of almost 7500 nt [11] and has a single open reading frame (ORF) encoding a polyprotein: structural P1 (viral protein (VP) 4, VP2, VP3, and VP1) and nonstructural proteins P2 (2A, 2B, and 2C) and P3 (3A, 3B, 3C, and 3D). The ORF is preceded by the 5′ untranslated region (UTR) and followed by the 3′ UTR with a poly(A) tail [12,13].

The HAV internal ribosomal entry site (IRES) exists in the 5′ UTR, and HAV initiates the translation by a cap-independent mechanism through the HAV IRES [14,15,16,17]. A single HAV polyprotein is proteolytically processed by the HAV 3C protease and cellular protease(s) into mature proteins. HAV 3D is an RNA-dependent polymerase that is important for viral replication. Highly permissive cell lines are needed to screen antivirals [18,19]. The potential antiviral targets are also important to develop effective antivirals. In this review article, we first discuss the cell culture systems. We next review the drug targets for HAV infections.

## 2. Cell Culture Systems for HAV Replication

### 2.1. Cell Lines Permissive for HAV Replication

HAV is a fastidious virus that grows slowly due to a combination of several factors [20]. HAV is difficult to grow, and only a few combinations of HAV strains and cell lines are available. Thus, it has been difficult to develop effective and cheap vaccines and to study potential antivirals.

Efficient infectious cell culture systems of HAV are shown in Table 1. Provost et al. successfully propagated HAV in primary marmoset hepatocytes and the normal fetal rhesus kidney cell line (FRhK6) [21]. They established methods of immunofluorescence, immunofluorescence blockade, serum neutralization, immune adherence, radioimmunoassay, immune electron microscopy, and marmoset inoculation tests for HAV detection [21]. Frösner et al. isolated HAV directly from human feces and propagated the virus serially in the human hepatoma cell line Alexander (PLC/PRF/5) [22]. They confirmed by radioimmunoassay that hepatitis A antigen (HAAg) increased in the cell extracts obtained by the freezing and thawing of cells [22].

Gauss-Müller et al. demonstrated that the antigen was located within the cytoplasm of HAV-infected human embryo fibroblasts by an immunofluorescence study [23]. Kojima et al. also propagated HAV in the conventional cell lines, FL and Vero cells [24]. They confirmed HAAg in cytoplasm by radioimmunoassay (RIA) and immune electron microscopy although they did not observe any cytopathic effects in the cell culture [24]. In primary African green monkey (*Cercopithecus aethiops*) kidney (AGMK) cell cultures, HAV strains were recovered from the stool specimen of a patient with HAV and confirmed by direct immunofluorescence [25].

Lemon et al. developed a new radioimmunofocus assay method, which retained many of the advantages of conventional plaque assays, for the quantitation of HAV using African green monkey kidney BS-C-1 cells [26]. Wheeler et al. reported that the HAS-15 strain, which was recovered from wild HAV, was adapted to rapid growth in FRhK-4 cells by more than 20 7-day passages, and confirmed HAV using a radioimmunoassay and virus-specific immunofluorescence [27].

Crance et al. reported that PLC/PRF/5 cells supported continuous production of the HAV CF53 strain, which was isolated from the stools of a patient with HAV infection 3 days after the onset of jaundice and was adapted to grow in PLC/PRF/5 cells [28]. Robertson et al. reported the growth and recovery of purified HAV from FRhK4 cells persistently infected with HAV isolate HAS-15 over a 2- to 3-month period [29]. Simian HAV strain AGM-27 and cell culture-adapted HM-175 grew in CV-1, FRhK-1, and primary AGMK cells [30].

Cohen et al. cloned the cDNA of a cell culture-adapted HAV (HAV HM-175/7 MK-5) full-length genome into pBR322 [31]. They transfected the cells using an infectious transcript from HAV cDNA into AGMK and CV-1 cells and inoculated with transfection-derived virus into marmosets and observed the appearance of anti-HAV antibodies and hepatitis [31]. Emerson et al. performed transfection of HM-175 (wild type and cell-culture-adapted) and an infectivity assay in FRhK-4 and AGMK cells and revealed that the attenuation of virulence may also require multiple mutations [32,33]. Morace et al. revealed that genome mutations of HAV 3A regions were associated with two cytopathic HAV strains [34]. The HAV strains FG and SI0 were isolated from the feces of a young patient from Southern Italy collected four days before the onset of clinical symptoms and from a water sample taken from the Tiber River (central Italy), respectively [34,35]. These sequences were compared to those of three cytopathic clones of HM175 described by Lemon et al. [36]. A study reported that HM175-cytopathogenic strains have mutations in both the 5′ and 3′ UTRs and in the nonstructural proteins 2A, 2B, 2C, 3A, and 3Dpol, which may be associated with the cytopathic phenotype [36]. Sequence analysis revealed cell culture-adapted HAV mutations [37,38,39,40].

The cynomolgus monkey renal cell line JTC-12.P3 could support HAV replication [41]. The pig cell line IB-RS-2 cells also supported HAV replication [42]. Mouse liver cells were coded for functional HAV receptors and other factors required for efficient HAV replication in cell culture [43].

Primary human hepatocyte PXB cells, which were derived from a humanized severe combined immunodeficiency albumin promoter/enhancer-driven urokinase-type plasminogen activator mouse model, could support HAV replication [44]. Cell culture-grown HAV strains and HAV derived from fecal extracts were used for the infection of these cell lines. Similar to the cell culture-grown HCV JFH1 strain [45,46,47], it may be easier to infect human hepatocytes with cell culture-grown HAV strains. Cytopathic effects are not always observed when these HAV strains infect human cell lines. Perez-Rodriguez et al. recently reported that quasispecies genomic selection and molecular breeding using deep sequencing identified high-fitness individuals improving HAV [48].

### 2.2. Cell Culture for HAV Vaccine Development

Inactivated vaccines are licensed and commonly used to control HAV infections [3]. HAV strain CR326, adapted to grow in *Macaca mulatta* kidney LLC-MK2 cells, was highly purified, inactivated with formalin, adsorbed to alum, and tested for the capacity to induce anti-HAV antibodies in both mice and marmosets [53]. As LLC-MK2 cells are unacceptable to prepare human vaccines, the HAV strain CR326 can also be prepared in a similar manner in Medical Research Council cell strain 5 (MRC-5) cells, which are diploid human cell lines composed of fibroblasts and acceptable for the manufacturing of human vaccines [53].

Flehming et al. performed HAV propagation and adaptation in human embryo kidney cells (HKC) [54]. They also demonstrated that HAV from the 10th passage through HKC can replicate in a human embryo fibroblast strain (HFS) [54]. They also developed the HAV strain HFS/GBM, which can be propagated in human fibroblast cells in quantities sufficient for producing inactivated vaccines [55]. These fibroblasts were derived from the lungs of a normal 25-week embryo.

As an alternative strain for a vaccine strain, a fast-growing strain of HAV with a great potential for HAV antigen production has been isolated by quasispecies genomic selection and molecular breeding [48]. As the production of vaccines is expensive [12], a fast-growing HAV strain may be useful for making the production costs of HAV vaccines lower.

### 2.3. Cell Culture for the Development of Anti-HAV Drugs

Despite the use of an effective vaccine, antivirals against HAV would be of great use [12]. The effective anti-HAV drugs and cell culture systems in which they were found are shown in Table 2. We previously reviewed other anti-HAV drugs (see the reference [13]). The human hepatoma cell lines PLC/PRF/5 and Huh7 are often used for the discovery of anti-HAV drugs, although HAV has various strains. HAV had high IRES activities in HLE and Huh7 cells [16]. Although excellent HAV vaccines exist, further development of therapeutic options to prevent severe hepatitis A is needed. In Japan, due to the legal difficulties associated with cadaveric donation that existed ~20 years ago, the number of liver transplantations is still lower than in other countries. Therefore, anti-HAV drugs must be developed. As most of the investigations did not go beyond tests on cell cultures, it would be useful and important to improve HAV cell culture systems.

## 3. HAV Subgenomic Replicon for the Study of Antiviral Drugs

The HAV subgenomic replicon has a luciferase reporter gene replacing nearly the entire P1 capsid region [66]. Stable expression of T7-promoted genes in cells either constitutively expressing T7 RNA polymerase or infected with a helper virus expressing T7 RNA polymerase can cause HAV subgenomic replicon RNA or cDNA replication in human cells [67,68,69].

Although we cannot evaluate the step of HAV infection in the HAV replicon system, HAV replication could be measured by a luciferase assay to evaluate effects of the drug more easily and safely than with live HAV [14,17,65,70]. We illustrated the structure of the HAV subgenomic replicon and HAV IRES–reporter constructs in comparison with the HAV full-length genome (see reference [17].)

## 4. Blocking the Entry Pathway as an Antiviral Strategy

Kaplan et al. reported that HAV cellular receptor 1 (HAVcr-1) is an attachment receptor for HAV as well as a functional receptor for HAV infection in African green monkey kidney (AGMK) GL37 cells [71]. Liver, as well as kidneys and colorectal tissues, have a high expression of HAVcr-1 mRNA [72]. HAVcr-1 is also known as kidney injury molecule-1 (KIM-1), T cell immunoglobulin (Ig), and mucin domain containing 1 (TIM-1). Major sequence variants of TIM-1 were completely co-segregated with T cells and the airway phenotype regulator (Tapr), which is associated with asthma susceptibility genes [73]. HAVcr-1 may also have an inverse relationship between HAV infection and atopy development [73]. Immunotherapies may regulate HAVcr-1 function and downmodulate allergic inflammatory diseases [74].

However, there are important discussions on the actual role of HAVcr-1 as a cell surface receptor for HAV [75,76,77]. Costafreda and Kaplan (2018) reported that AGMK HAVcr-1 knockout cells lost susceptibility to HAV infection, including HAV-free viral particles and exosomes purified from HAV-infected cells [76]. Naked HAV virions were responsible for fecal–oral transmission of HAV between individuals and eHAV mediated the spread within the newly infected host [10,75].

In contrast, in other studies, the HAVcr-1 was identified as a non-essential entry factor for either naked HAV or eHAV [77,78]. Integrin β1 binds elsewhere on the HAV capsid. Trafficking of eHAV to the lysosome is essential for the entry and uncoating of HAV genome and requires the endosomal sorting complexes required for transport (ESCRT) and programmed cell death 6-interacting protein/apoptosis-linked gene 2 (ALG2)-interacting protein X (PDCD6IP/ALIX) in addition to RAB5A, the member RAS oncogene family (Rab5), and RAB7, the member RAS oncogene family (Rab7) GTPase [75]. The present proposed life cycle model of HAV is shown in Figure 1. Although the mechanism of endocytosis has been developed in detail, the involvement of HAV and cell surface receptors for HAV is not clearly established yet. Drugs inhibiting the steps of entry process could be efficient antivirals.

## 5. Inhibiting the HAV IRES-Mediated Translation in Human Hepatoma Cell Lines

HAV translation is initiated in a cap-independent style by a mechanism involving the binding of the 40S RNA subunit at a portion located hundreds of bases downstream of the 5′ end of the HAV RNA genome, which has been termed “HAV IRES”. HAV IRES spans a region from ~nt.161 to the first initiator codon AUG, located at ~nt. 734, and encompasses most of the 5′ UTR of HAV RNA [79]. We previously reported that some siRNAs targeting the HAV IRES suppressed the genome translation and replication [14].

As the nucleotide sequence homology of the 5′ end of the HAV RNA genome is well-preserved among different HAV strains, HAV IRES may be used as a universal and effective target for anti-HAV drugs. HAV IRES is highly ineffective [83,84]; however, mutants with higher activity have been described [20,48], which may be useful for the study of antivirals. Additionally, HAV IRES is much more active in the FRhK-4 cells than in the Huh7 cells. However, the Huh7 cells represent a better model of the natural infection and a closer model for testing antivirals.

As HAV had higher IRES activities in Huh7 cells [16], researchers typically used Huh7 to examine HAV IRES activity with transient transfection of a bicistronic reporter construct containing HAV IRES as an intragenic spacer (pSV40 HAV IRES). After 48–72 h of transfection and treatment with or without specific drugs, the activities of HAV IRES were measured using a luciferase assay. Studies demonstrated that amantadine inhibited HAV IRES-mediated translation in human hepatoma cells [15,16,17]. Researchers also used the stable cell line, COS-7 HAV IRES, and found that the JAK2 inhibitor AZD1480 inhibited HAV replication [63]. HAV IRES was an attractive target for anti-HAV drugs and could be useful for screening effective anti-HAV drugs (Figure 1).

## 6. HAV 3C Protease and 3D Polymerase May Be other Candidates for Anti-HAV Drug Targets

In the treatment of chronic hepatitis C infection, HCV NS3/4A protease inhibitors and HCV NS5B nucleotide inhibitors now play major roles in achieving a sustained virological response (SVR) [85,86]. HAV 3C is a cysteine protease inhibitor and processes HAV proteins [13]. HAV 3C protease can impair the induction of β interferon through the cleavage of the NF-κB essential modulator (NEMO) and plays an immune role in the evasion mechanism of HAV [87]. HAV 3C also induced caspase-independent cell death with the vacuolization of lysosomal and endosomal organelles [88]. There are several reports on the development of HAV 3C inhibitors [13,81].

HAV 3D is an RNA-dependent RNA polymerase and plays a role in the progression of acute hepatitis A. Amino acid changes in HAV 3D could modulate the growth rate of HAV [82]. HAV 3D was also involved in the disruption of TLR3 signaling, an important pathway for HAV eradication [89]. Although there are a few reports regarding HAV 3D polymerase inhibitors, HAV 3D could be a potential target for the treatment of hepatitis A.

## 7. Conclusions

In the present article, we described the relationship between anti-HAV drugs and cell culture models for HAV infection, HAV subgenomic replicons, and HAV IRES reporter assays. Creating better and cheaper HAV vaccines and developing anti-HAV drugs against severe hepatitis A remains important and further exploration is needed.

## Figures and Tables

**Figure 1 viruses-12-00533-f001:**
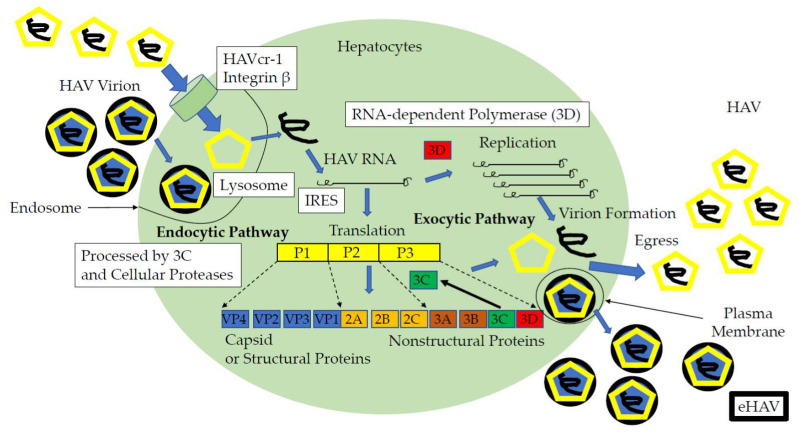
Life cycle of the hepatitis A virus and candidates of anti-HAV drug targets (open squares). HAVcr-1, hepatitis A virus cellular receptor 1; IRES, internal ribosomal entry site; HAV, naked HAV; eHAV, quasi-enveloped HAV; bold black circle, envelope; yellow pentagon, viral capsid [10,13,71,72,73,74,75,76,77,78,79,80,81,82].

**Table 1 viruses-12-00533-t001:** Cell lines supporting hepatitis A virus (HAV) replication.

Authors (Year) [References]	Cell Lines	HAV Strain	Duration of HAV Infection	Cytopathy
Provost, P.J., et al. (1979) [21]	Primary marmoset liver and fetal rhesus kidney (FRhK6) cells	Marmoset-adapted CR326	NA	No
Frösner, G.G., et al. (1979) [22]	Alexander (PLC/PRF/5) cells	MS-1	7 weeks	No
Gauss-Müller, V., et al. (1981) [23]	Human embryo fibroblasts	Cell culture-adapted strain	90 and 210 days	No
Kojima, H., et al. (1981) [24]	FL or Vero cells	HAV derived from fecal extracts	18 days	No
Daemer, R.J., et al. (1981) [25]	Primary African green monkey kidney (AGMK) cells	MS-1, SD-1, HM-175	11 weeks	No
Lemon, S.M., et al. (1983) [26]	BS-C-1 cells	HM-175, PA-21	30 days	No
Wheeler, C.M., et al. (1986) [27]	FRhK-4 cells	HAS-15	20 × 7 passages	No
Cohen, J.I., et al. (1987) [31]	AGMK or CV-1 cells	HAV cDNA HM-175n MK-5	5 weeks	No
Crance, J.M., et al. (1987) [28]	PLC/PRF/5 cells	CF53	6–12 months	No
Robertson, B.H., et al. (1988) [29]	FRhK-4 cells	HAS-15	2–3 months	No
Tsarev, S.A., et al. (1991) [30]	Primary AGMK or FRhK-4 cells	AGM-27	14 days	No
Emerson, S.U., et al. (1992) [32]	FRhK-4 cells	HM175	2 months	No
Emerson, S.U., et al. (1992) [33]	AGMK cells	HM175	120 days	No
Morace, G., et al. (1993) [34]	Frp/3 cells	HM175 cytopathic clone	7–9 days	Yes
Baba, M., et al. (1993) [41]	JTC-12.P3 cells	HAV	8 weeks	No
Graff, J., et al. (1994) [37]	FRhK-4 or HFS cells	GBM/WT, GBM/FRhK, GBM/HFS	14 days	No
Dotzauer, A., et al. (1994) [42]	GPE or SP 1K cells	HM175	42 days	No
Graff, J., et al. (1994) [38]	FRhK-4 cells	GBM/Fp2	60 days	No
Zhang, H., et al. (1995) [39]	BS-C-1 cells	HM175/18f	14 days	Yes
Funkhouser, A.W., et al. (1996) [40]	MRC-5 cells	MR8 or MRC-5/9	160 days	No
Feigelstock, D.A., et al. (2005) [43]	GL37 or MMH-D3 cells	HM175	14–50 days	No
Konduru, K., et al. (2006) [49]	Huh7-A-I cells	WT HM175	16 days	No
Kusov, Y., et al. (2006) [50]	Huh7 cells	Huh-7/HAV	14 days	No
Jiang, X., et al. (2014) [51]	Huh7 or GL37 cells	HA11-1299 GT IIIA or KRM003 GT IIIB	4 days	No
Hirai-Yuki, A., et al. (2016) [52]	Caco-2 or HepG2-N cells	HM175/p16	4–7 days	No
Pérez-Rodríguez, F.J., et al. (2016) [48]	FRhK-4	HM175-HP, F0.05LA	7 days	No
Win, N.N., et al. (2018) [44]	PXB cells	HA11-1299 GT IIIA	7 days	No

NA, not available; WT, wild type; GT, genotype.

**Table 2 viruses-12-00533-t002:** Effective drugs inhibiting hepatitis A virus (HAV) replication discovered in cell culture systems for HAV.

Authors (Year) [References]	Cell Lines	HAV Strain	Effective Anti-HAV Drugs
Widell, A., et al. (1986) [56]	Frhk-4	H 141	Arabinosylcytosine, amantadine, ribavirin
Biziagos, E., et al. (1987) [57]	PLC/PRF/5	CF53	Taxifolin, atropine
Biziagos, E., et al. (1990) [58]	PLC/PRF/5	CF53	Atropine, protamine, atropine/protamine combination
Crance, J.M., et al. (1990) [59]	PLC/PRF/5	CF53	Ribavirin, amantadine, pyrazofurin, glycyrrhizin
Girond, S., et al. (1991) [60]	PLC/PRF/5	CF53	Sulphated polysaccharides
Crance, J.M., et al. (1995) [61]	PLC/PRF/5	CF53	Interferon-alpha
Kusov Y., et al. (2006) [50]	Huh7	Huh-7/HAV	siRNA
Yang, L., et al. (2010) [17]	GL37	KRM003	Amantadine, Interferon-alpha
Kanda, T., et al. (2010) [62]	GL37	KRM003	Interferon-lambda
Jiang, X., et al. (2015) [63]	Huh7	HA11-1299	AZD1480
Kanda, T., et al. (2015) [64]	Huh7	HA11-1299	Sirtinol
Win, N.N., et al. (2019) [44]	Huh7 PXB	HA11-1299	Japanese rice koji miso extracts
Ogawa, M., et al. (2019) [65]	Huh7	HA11-1299	Zinc sulfate

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
