# Peer review of "Cell Culture Systems and Drug Targets for Hepatitis A Virus Infection"

_viruses, 2020, doi:10.3390/v12050533_

Round 1

Reviewer 1 Report

This ms intends to be a review on cell culture systems and drug targets for hepatitis A Virus infection. The idea for such a review is interesting but still not achieved in the present form.

General Comments

  • The ms does not make clear the relationship between the culture systems, available only for a few cell culture adapted HAV strains, and the antiviral drug targets. The authors should find a guiding line to go from the permissive cells to the potential antiviral targets.
  • There is a highly relevant and missing point: HAV is a fastidious virus to grow due to a combination of factors (Pintó et al. 2018. Cold Spring Harbor Perspectives in Medicine. 2018. 8: 71-81; pii: a031781). Once this point become clear, one could understand the difficulties for the vaccine production and for the study of potential antivirals. Thus, it should be stated from the very beginning that HAV is very difficult to grow and that only a few strains/cell combinations exist. In this regard some references are missing in the 2.1 section (Pérez-Rodríguez et al. 2016. Scientific Reports 6: 35962).
  • In section 2.1. there is an explanation of the schedule of vaccination for the Japanese inactivated vaccine based on the genotype IIIB KRM003 strain. This point is out of scope and in fact, the schedule for the rest of vaccines is not given. Thus, lines 71-75 should be removed, or alternatively information on other of the most common vaccines should be added.  
  • The position of sections 4 and 5 should be interchanged. Following the biological cycle the interaction with receptor comes first, followed by the RNA uncoating and the IRES-directed translation.
  • Indeed, the IRES could be a good target for antivirals. The IRES of HAV is highly ineffective (Brown et al. 1991. Journal of Virology65: 5828; Whetter et al. 1994. Journal of Virology 68: 5253), but mutants with higher activity have been described (Pérez-Rodríguez et al. 2016. Scientific Reports 6: 35962; Pintó et al. 2018. Cold Spring Harbor Perspectives in Medicine. 2018. 8: 71-81; pii: a031781), which maybe very useful for the study of antivirals. Additionally, the HAV IRES is much more active in the FRhK-4 cells than in the Huh7 cells. However, these latter cells represent a better model of the natural infection and thus a closer model for testing antivirals.
  • Regarding the HAVcR1 receptor, there is presently an important discussion on its actual role (Rivera-Serrano et al. 2019. eLife 8: e43983. Costafred a & Kaplan. 2018. Journal of Virology 92: e02065-17 Das et al. 2019. Journal of Virology 93: e01793-18), that should be taken into consideration in section 5.

Specific Comments

  • Lines 24-25. Remove worldwide.
  • Line 27. Remove virus
  • Line 28. Remove the entire sentence.
  • Line 27. Move this sentence to line 25.
  • Line 31. Remove human.
  • In line 46, what is the meaning of “cell lines Huh7 and PLC/PRF/5 can support HAV replication for up to 12 months”? Are the authors referring to persistent infections? Similarly, in lines 52-53, what does it mean support replication for 50 days? Do the authors mean that the virus was produced for 50 days of passing the persistently infected cells? And so on thereafter.
  • Line 49. Adopted should be adapted
  • In Table 1 an explanation of the meaning of the detection of HAV is required. This reviewer can only understand the content in relation to persistently infected cells. If this is the case, information on cytopathogenic strains (i.e. pHM175 and derived strains) should be provided. For instance, a fast-growing strain has been recently isolated which produces very large plaques and high titers of viruses in the supernatant of FRhk-4 cells (Pérez-Rodríguez et al. 2016. Scientific Reports 6: 35962).
  • Line 50. Again what is the meaning for “at least one week”?
  • Line 70. Thelung needs a space.
  • Line 80. This sentence is difficult to be understood.
  • Section 3 seems unnecessary in the context of this review.
  • Lines 85-88. Not necessary this introduction on PV1.
  • Line 90. Remove the last HuH7 cells.
  • Line 91. Meaning of this sentence?
  • Remove lines 127-130. Since are out of the scope of this review.
  • Figure 1 is too simple and does not fit with the present knowledge on the HAV cycle (the receptor is still a controversial issue, the uncoating takes place in the endocytic pathway, the virus egress uses the exocytic pathway in the form of exosomes,…).

Author Response

Response to the Reviewer 1: Thank you for your encouraging comments and criticisms.

Response to your General Comments: “The ms does not make clear the relationship between the culture systems, available only for a few cell culture adapted HAV strains, and the antiviral drug targets. The authors should find a guiding line to go from the permissive cells to the potential antiviral targets.”

Thank you for your valuable comments. We agree with you. We revised our manuscript as follows.

In lines 39-49,

…. important for viral replication. Highly permissive cell lines are needed to screen antivirals [18,19]. The potential antiviral targets are also important to develop effective antivirals. In this review article, we first discuss the cell culture systems. We next review the drug targets for HAV infections.

  1. Cell culture systems for HAV replication

2.1. Cell lines permissive for HAV replication

HAV is a fastidious virus that grows slowly due to a combination of several factors [20]. HAV is difficult to grow, and only a few combinations of HAV strains and cell lines are available. Thus, it has been difficult to develop effective and cheaper vaccines and to study potential antivirals. Perez-Rodriguez et al. recently reported that quasispecies genomic selection and molecular breeding using deep-sequencing identified high-fitness improving HAV [21].…..

Response to your General Comments: “There is a highly relevant and missing point: HAV is a fastidious virus to grow due to a combination of factors (Pintó et al. 2018. Cold Spring Harbor Perspectives in Medicine. 2018. 8: 71-81; pii: a031781). Once this point become clear, one could understand the difficulties for the vaccine production and for the study of potential antivirals. Thus, it should be stated from the very beginning that HAV is very difficult to grow and that only a few strains/cell combinations exist. In this regard some references are missing in the 2.1 section (Pérez-Rodríguez et al. 2016. Scientific Reports 6: 35962).”

Thank you for your valuable comments. We agree with you. We added new references [20] and [21], and revised our manuscript as follows.

In lines 39-49,

…. important for viral replication. Highly permissive cell lines are needed to screen antivirals [18,19]. The potential antiviral targets are also important to develop effective antivirals. In this review article, we first discuss the cell culture systems. We next review the drug targets for HAV infections.

  1. Cell culture systems for HAV replication

2.1. Cell lines permissive for HAV replication

HAV is a fastidious virus that grows slowly due to a combination of several factors [20]. HAV is difficult to grow, and only a few combinations of HAV strains and cell lines are available. Thus, it has been difficult to develop effective and cheaper vaccines and to study potential antivirals. Perez-Rodriguez et al. recently reported that quasispecies genomic selection and molecular breeding using deep-sequencing identified high-fitness improving HAV [21].…..

Response to your General Comments: “In section 2.1. there is an explanation of the schedule of vaccination for the Japanese inactivated vaccine based on the genotype IIIB KRM003 strain. This point is out of scope and in fact, the schedule for the rest of vaccines is not given. Thus, lines 71-75 should be removed, or alternatively information on other of the most common vaccines should be added.”

Thank you for your valuable comments. We agree with you. We deleted this description.

Response to your General Comments: “The position of sections 4 and 5 should be interchanged. Following the biological cycle the interaction with receptor comes first, followed by the RNA uncoating and the IRES-directed translation.”

Thank you for your valuable comments. We agree with you. We revised our manuscript accordingly.

Response to your General Comments: “Indeed, the IRES could be a good target for antivirals. The IRES of HAV is highly ineffective (Brown et al. 1991. Journal of Virology65: 5828; Whetter et al. 1994. Journal of Virology 68: 5253), but mutants with higher activity have been described (Pérez-Rodríguez et al. 2016. Scientific Reports 6: 35962; Pintó et al. 2018. Cold Spring Harbor Perspectives in Medicine. 2018. 8: 71-81; pii: a031781), which maybe very useful for the study of antivirals. Additionally, the HAV IRES is much more active in the FRhK-4 cells than in the Huh7 cells. However, these latter cells represent a better model of the natural infection and thus a closer model for testing antivirals.”

Thank you for your valuable comments. We agree with you. We revised our manuscript accordingly.

In lines 180-183,

…HAV drugs. HAV IRES is highly ineffective [83,84]; however, mutants with higher activity have been described [[20,21], which may be useful for the study of antivirals. Additionally, HAV IRES is much more active in the FRhK-4 cells than in the Huh7 cells. However, the Huh7 cells represent a better model of the natural infection and a closer model for testing antivirals.

Response to your General Comments: “Regarding the HAVcR1 receptor, there is presently an important discussion on its actual role (Rivera-Serrano et al. 2019. eLife 8: e43983. Costafred a & Kaplan. 2018. Journal of Virology 92: e02065-17 Das et al. 2019. Journal of Virology 93: e01793-18), that should be taken into consideration in section 5.”

Thank you for your valuable comments. We agree with you. We revised our manuscript accordingly.

In lines 147-169,

Kaplan et al. reported that the HAV cellular receptor 1 (HAVcr-1) is an attachment receptor for HAV as well as a functional receptor for HAV infection in African Green Monkey kidney (AGMK) GL37 cells [72]. Liver, as well as kidney and colorectal tissues, have a high expression of HAVcr-1 mRNA [76]. HAVcr-1 is also known as kidney injury molecule-1 (KIM-1), T-cell immunoglobulin (Ig), and mucin domain-containing 1 (TIM-1). Major sequence variants of TIM-1 were completely co-segregated with T cells and the airway phenotype regulator (Tapr), which is associated with asthma susceptibility genes [77]. HAVcr-1/Tim-1 may also have an inverse relationship between HAV infection and atopy development [77]. Immunotherapies may regulate HAVcr-1/TIM-1 function and downmodulate allergic inflammatory diseases [78]. The life cycle of HAV is shown in Figure 1.

There are important discussions on the actual role of HAVcr-1 [79-81]. Costafreda and Kaplan (2018) reported that AGMK HAVcr-1 knockout cells lost susceptibility to HAV infection, including HAV-free viral particles and exosomes purified from HAV-infected cells [79]. Naked HAV virions were responsible for fecal-oral transmission of HAV between individuals and eHAV mediated the spread within the newly infected host [10; 80].

HAVcr-1 is an important attachment factor for eHAV in GL37 cells, although HAVcr-1 is not an essential entry factor for either naked HAV or eHAV [81,82]. Integrin β1 binds elsewhere on the HAV capsid. Trafficking of eHAV to the lysosome is essential for the entry and uncoating of HAV genome and requires the endosomal-sorting complexes required for transport (ESCRT) and programmed cell death 6 interacting protein (PDCD6IP/ALIX), in addition to RAB5A, the member RAS oncogene family (Rab5) and RAB7, member RAS oncogene family (Rab7) GTPase [80]. However, HAVcr-1, HAV IRES, host cellular and HAV 3C protease, and HAV 3D polymerase are also candidates for anti-HAV drug targets (Figure 1) [13, 72-75].

Response to your Specific Comments: “Lines 24-25. Remove worldwide.”

Thank you for your valuable comments. We agree with you. We revised our manuscript accordingly.

Response to your Specific Comments: “Line 27. Remove virus”

Thank you for your valuable comments. We agree with you. We revised this part.

Response to your Specific Comments: “Line 28. Remove the entire sentence.”

Thank you for your valuable comments. We agree with you. We revised our manuscript accordingly.

Response to your Specific Comments: “Line 27. Move this sentence to line 25.”

Thank you for your valuable comments. We agree with you. We revised our manuscript accordingly.

Response to your Specific Comments: “Line 31. Remove human.”

Thank you for your valuable comments. We agree with you. We revised our manuscript accordingly.

Response to your Specific Comments: “In line 46, what is the meaning of “cell lines Huh7 and PLC/PRF/5 can support HAV replication for up to 12 months”? Are the authors referring to persistent infections? Similarly, in lines 52-53, what does it mean support replication for 50 days? Do the authors mean that the virus was produced for 50 days of passing the persistently infected cells? And so on thereafter.”

Thank you for your valuable comments. We agree with you. We deleted these parts.

Response to your Specific Comments: “Line 49. Adopted should be adapted”

Thank you for your valuable comments. We agree with you. We revised our manuscript accordingly.

Response to your Specific Comments: “In Table 1 an explanation of the meaning of the detection of HAV is required. This reviewer can only understand the content in relation to persistently infected cells. If this is the case, information on cytopathogenic strains (i.e. pHM175 and derived strains) should be provided. For instance, a fast-growing strain has been recently isolated which produces very large plaques and high titers of viruses in the supernatant of FRhk-4 cells (Pérez-Rodríguez et al. 2016. Scientific Reports 6: 35962).”

Thank you for your valuable comments. We agree with you. We revised Table 1 and our manuscript accordingly.

In lines 84-92,

…. revealed that the attenuation of virulence may also require multiple mutations [33,34]. Morace et al. revealed that genome mutations of HAV 3A regions were associated with two cytopathic HAV strains [35]. The HAV strains FG and SI0 were isolated from the feces of a young patient from Southern Italy, collected four days before the onset of clinical symptoms and from a water sample taken from the Tiber River (central Italy), respectively [35, 48]. These sequences were compared to those of three cytopathic clones of HM175 described by Lemon et al. [49]. A study reported that HM175-cytopathogenic strains have mutations in both the 5' and 3' UTRs and in the nonstructural proteins 2A, 2B, 2C, 3A, and 3Dpol, which may be associated with the cytopathic phenotype [49]. Sequence analysis revealed the cell culture-adapted HAV mutations [37,39-41].

Response to your Specific Comments: “Line 50. Again what is the meaning for “at least one week”?”

Thank you for your valuable comments. We agree with you. We deleted this sentence.

Response to your Specific Comments: “Line 70. Thelung needs a space.”

Thank you for your valuable comments. We agree with you. We revised our manuscript accordingly.

Response to your Specific Comments: “Line 80. This sentence is difficult to be understood.”

Thank you for your valuable comments. We agree with you. We revised our manuscript accordingly.

Response to your Specific Comments: “Section 3 seems unnecessary in the context of this review.”

Thank you for your valuable comments. We agree with you. We revised our manuscript according to the comments of Reviewer 2.

Response to your Specific Comments: “Lines 85-88. Not necessary this introduction on PV1.”

Thank you for your valuable comments. We agree with you. We deleted this part.

Response to your Specific Comments: “Line 90. Remove the last HuH7 cells.”

Thank you for your valuable comments. We agree with you. We revised our manuscript accordingly.

Response to your Specific Comments: “Line 91. Meaning of this sentence?”

Thank you for your valuable comments. We agree with you. We deleted this sentence.

Response to your Specific Comments: “Remove lines 127-130. Since are out of the scope of this review.”

Thank you for your valuable comments. We agree with you. We deleted these sentences.

Response to your Specific Comments: “Figure 1 is too simple and does not fit with the present knowledge on the HAV cycle (the receptor is still a controversial issue, the uncoating takes place in the endocytic pathway, the virus egress uses the exocytic pathway in the form of exosomes,…).”

Thank you for your valuable comments. We agree with you. We revised Figure 1.

Reviewer 2 Report

The review written by Kanda et al aims to provide an overview of cell culture systems for the study of the replication cycle of the Hepatitis A virus and the identification of potential targets for treatment.

This manuscript undeniably represents a good contribution to the field since, to my knowledge, there is no review that lists all the available cell models.  Nevertheless, I think that, in its current form, the manuscript suffers from some structural awkwardness that prevents it from providing a clear message.

Below is a list of my comments/suggestions:

6- By reading the manuscript, one gets the impression that it was constructed as two independent reviews, one dealing with models and the other with targets for drugs, with no real link between the two.  Clear connections between these two topics should be made in order to make the reading more fluid.

2- the review should focus only on HAV and the paragraph on other viruses (lines 128-132) should be omitted since it does not bring anything to the point. 

3- Table 3, as well as the text referring to it, adds nothing and could be removed without altering the message. 

4- On the contrary, some paragraphs (which really fit into the topic of the review) should be completed. For example, there is a gap between the detailed information given in table 1 and the text describing it.  Many cell lines and viral strains that are listed in the table are not commented at all in the text (page 2).  More detailed text should be provided as this part is one of the main purposes of the review.

5- the paragraph concerning cell culture models used for vaccine development should be reorganized.  In its current form, it is not clear to the reader which are the historical models and which are currently used.  Therefore, this chapter could be divided into two sub-paragraphs, one dealing with the history of vaccine development and the other with currently available vaccines. In addition, some information is missing, such as the infection of MRC-5 cells with the HM175 strain from which the HAVRIX vaccine is produced. A table could be added to this chapter to list all vaccines available on the market and indicate where and how they are used.

6- Paragraph 2.1 concerning anti-HAV drugs is very short and lacks much important information.  First of all, the way it is written leaves the impression that all the drugs listed in Table 2 are effective and used clinically, which is absolutely not the case.  The authors should therefore make it clearer that most of the investigations did not go beyond tests on cultured cells.  In addition, the rationale for developing drugs should be better stated. 

7- the paragraph on subgenomic replicon of poliovirus is out of context and should be removed.

8- a figure showing the different reporter constructs (subgenomic, HAV IRES construct, in comparison with the full length genome) should be included.

Author Response

Response to the Reviewer 2: Thank you for your encouraging comments and criticisms.

Response to your Comments: “By reading the manuscript, one gets the impression that it was constructed as two independent reviews, one dealing with models and the other with targets for drugs, with no real link between the two.  Clear connections between these two topics should be made in order to make the reading more fluid.”

Thank you for your valuable comments. We agree with you. We revised our manuscript as follows.

In lines 38-48,

…. important for viral replication. Highly permissive cell lines are needed to screen antivirals [18,19]. The potential antiviral targets are also important to develop effective antivirals. In this review article, we first discuss the cell culture systems. We next review the drug targets for HAV infections.

  1. Cell culture systems for HAV replication

2.1. Cell lines permissive for HAV replication

HAV is a fastidious virus that grows slowly due to a combination of several factors [20]. HAV is difficult to grow, and only a few combinations of HAV strains and cell lines are available. Thus, it has been difficult to develop effective and cheaper vaccines and to study potential antivirals. Perez-Rodriguez et al. recently reported that quasispecies genomic selection and molecular breeding using deep-sequencing identified high-fitness improving HAV [21].…..

Response to your Comments: “the review should focus only on HAV and the paragraph on other viruses (lines 128-132) should be omitted since it does not bring anything to the point.”

Thank you for your valuable comments. We agree with you. We omitted this part.

Response to your Comments: “Table 3, as well as the text referring to it, adds nothing and could be removed without altering the message.”

Thank you for your valuable comments. We agree with you. We omitted Table 3 and revised our manuscript accordingly.

Response to your Comments: “On the contrary, some paragraphs (which really fit into the topic of the review) should be completed. For example, there is a gap between the detailed information given in table 1 and the text describing it.  Many cell lines and viral strains that are listed in the table are not commented at all in the text (page 2). More detailed text should be provided as this part is one of the main purposes of the review.”

Thank you for your valuable comments. We agree with you. We revised our manuscript as follows.

In lines 45-92,

2.1. Cell lines permissive for HAV replication

HAV is a fastidious virus that grows slowly due to a combination of several factors [20]. HAV is difficult to grow, and only a few combinations of HAV strains and cell lines are available. Thus, it has been difficult to develop effective and cheaper vaccines and to study potential antivirals. Perez-Rodriguez et al. recently reported that quasispecies genomic selection and molecular breeding using deep-sequencing identified high-fitness improving HAV [21].

Efficient infectious cell culture systems of HAV are shown in Table 1 [22-47]. Provost et al. successfully propagated HAV in primary marmoset hepatocytes and the normal fetal rhesus kidney cell line (FRhK6) [22]. They established methods of immunofluorescence, immunofluorescence blockade, serum neutralization, immune adherence, radioimmunoassay, immune electron microscopy, and marmoset inoculation tests for HAV detection [22]. Frösner et al. isolated HAV directly from human feces and propagated the virus serially in the human hepatoma cell line Alexander (PLC/PRF/5) [23]. They confirmed by radioimmunoassay that hepatitis A antigen (HAAg) increased in the cell extracts obtained by the freezing and thawing of cells [23].

Gauss-Müller et al. demonstrated that the antigen was located within the cytoplasm of HAV infected human embryo fibroblasts by an immunofluorescence study [24]. Kojima et al. also propagated HAV in the conventional cell lines, FL and Vero cells [25]. They confirmed HAAg in cytoplasma by radioimmunoassay (RIA) and immune electron microscopy although they did not observe any cytopathic effects in the cell culture [25]. In primary African Green Monkey (Cercopithecus aethiops) kidney (AGMK) cell cultures, HAV strains were recovered from the stool specimen of a patient with HAV and confirmed by direct immunofluorescence [26].

Lemon et al. developed a new radioimmunofocusassay method, which retained many of the advantages of conventional plaque assays, for the quantitation of HAV using African Green Monkey kidney BS-C-1 cells [27]. Wheeler et al. reported that the HAS-15 strain, which was recovered from wild HAV, was adapted to rapid growth in FRhK-4 cells by more than 20 7-day passages, and confirmed HAV using a radioimmunoassay and virus-specific immunofluorescence [28].

Crance et al. reported that PLC/PRF/5 cells supported continuous production of the HAV CF53 strain, which was isolated from the stools of a patient with HAV infection 3 days after the onset of jaundice and was adapted to grow in PLC/PRF/5 cells [30]. Robertson et al. reported the growth and recovery of purified HAV from FRhK4 cells persistently infected with HAV isolate HAS-15 over a 2 to 3 month period [31]. Simian HAV strain AGM-27 and cell culture-adapted HM-175 grew in CV-1, FRhK-1, and primary AGMK cells [32].

Cohen et al. cloned the cDNA of a cell culture-adapted HAV (HAV HM-175/7 MK-5) full-length genome into pBR322 [29]. They transfected using an infectious transcript from HAV cDNA into AGMK and CV-1 cells and inoculated with transfection-derived virus into marmosets and observed the appearance of anti-HAV antibodies and hepatitis [29]. Emerson et al. performed transfection of HM-175 (wild type and cell-culture-adapted) and an infectivity assay in FRhK-4 and AGMK cells and revealed that the attenuation of virulence may also require multiple mutations [33,34]. Morace et al. revealed that genome mutations of HAV 3A regions were associated with two cytopathic HAV strains [35]. The HAV strains FG and SI0 were isolated from the feces of a young patient from Southern Italy, collected four days before the onset of clinical symptoms and from a water sample taken from the Tiber River (central Italy), respectively [35, 48]. These sequences were compared to those of three cytopathic clones of HM175 described by Lemon et al. [49]. A study reported that HM175-cytopathogenic strains have mutations in both the 5' and 3' UTRs and in the nonstructural proteins 2A, 2B, 2C, 3A, and 3Dpol, which may be associated with the cytopathic phenotype [49]. Sequence analysis revealed the cell culture-adapted HAV mutations [37,39-41].…

Response to your Comments: “the paragraph concerning cell culture models used for vaccine development should be reorganized.  In its current form, it is not clear to the reader which are the historical models and which are currently used.  Therefore, this chapter could be divided into two sub-paragraphs, one dealing with the history of vaccine development and the other with currently available vaccines. In addition, some information is missing, such as the infection of MRC-5 cells with the HM175 strain from which the HAVRIX vaccine is produced. A table could be added to this chapter to list all vaccines available on the market and indicate where and how they are used.”

Thank you for your valuable comments. We agree with you. We deleted the description of vaccine, according to Reviewer 1.

Response to your Comments: “Paragraph 2.1 concerning anti-HAV drugs is very short and lacks much important information.  First of all, the way it is written leaves the impression that all the drugs listed in Table 2 are effective and used clinically, which is absolutely not the case.  The authors should therefore make it clearer that most of the investigations did not go beyond tests on cultured cells.  In addition, the rationale for developing drugs should be better stated.”

Thank you for your valuable comments. We agree with you. But we have already reported this matter in Ref. [13]. We revised our manuscript as follows.

In lines 119-129,

The effective anti-HAV drugs and cell culture systems in which they were found are shown in Table 2. We previously reviewed other anti-HAV drugs (see the reference [13]). The human hepatoma cell lines PLC/PRF/5 and Huh7 are often used for the discovery of anti-HAV drugs, although HAV has various strains. HAV had high IRES activities in HLE and Huh7 cells [16]. Although excellent HAV vaccines exist, further development of therapeutic options to prevent severe hepatitis A is needed. In Japan, due to the legal difficulties associated with cadaveric donation that existed ~20 years ago, the number of liver transplantations is still less in Japan than in the other countries. Therefore, anti-HAV drugs must be developed. As most of the investigations did not go beyond tests on cell cultures, it would be useful and important to improve HAV cell culture systems. Quasispecies genomic selection and molecular breeding of HAV may improve virus production [21].…

Response to your Comments: “the paragraph on subgenomic replicon of poliovirus is out of context and should be removed.”

Thank you for your valuable comments. We agree with you. We deleted the paragraph on subgenomic replicon of poliovirus.

Response to your Comments: “a figure showing the different reporter constructs (subgenomic, HAV IRES construct, in comparison with the full length genome) should be included.”

Thank you for your valuable comments. We agree with you. But we have already reported this matter in Ref. [17]. We revised our manuscript as follows.

In lines 139-140,

…than live HAV [14,17,65,70]. We illustrated the structure of the HAV subgenomic replicon and HAV IRES-reporter constructs, in comparison with the HAV full length genome (see the reference [17].)

Round 2

Reviewer 1 Report

Specific point to improve the ms:

Lines 48-49. This sentence is out of context at this point. It should be moved at the end of the section 2.1. and be included in Table 1.

Line 106. “Inactivated vaccines are commonly used for HAV infections”. Replaced by “Inactivated vaccines are commonly used to control HAV infections”.

Lines 128-129. This sentence would be most adequate as "an alternative strain for a vaccine production" and thus moved to section 2.2. in a context such as “A fast growing strain of HAV with a great potential for antigen production has been isolated by quasispecies genomic selection and molecular breeding [21]”.

Line 132. Section 3. HAV subgenomic replicon, should be replaced by “HAV subgenomic replicon for the study of antiviral dugs”.

Line 146. This section should read “Blocking the entry pathway as an antiviral strategy”

Line 157. Add, the word however, at the beginning of the sentence.

Lines 162-163. The sentence “HAVcr-1 is an important attachment factor for eHAV in GL37 cells, although HAVcr-1 is not an 162 essential entry factor for either naked HAV or eHAV [81,82]”, should be better read “In contrast, in other studies the HAVcr-1 is identified as a non-essential entry factor for either naked HAV or eHAV [81,82]”.

Lines 168. Add the sentence "Drugs inhibiting the steps of the entry process could be efficient antivirals".

Lines167-168. Remove the sentence “However, HAVcr-1, 167 HAV IRES, host cellular and HAV 3C protease, and HAV 3D polymerase are also candidates for anti-168 HAV drug targets (Figure 1) [13, 72-75]”.

Line 171. Section 5. “Evaluation of HAV IRES-mediated translation in human hepatoma cells”, should read “Inhibiting the HAV IRES-mediated translation in human hepatoma cells”.

Author Response

Response to the Reviewer 1: Thank you for your encouraging comments and criticisms.

Response to your specific comment 1: “Lines 48-49. This sentence is out of context at this point. It should be moved at the end of the section 2.1. and be included in Table 1.”

Thank you for your valuable comments. We agree with you. According to your suggestion, we revised our manuscript as follows.

In lines 97-99,

…. Cytopathic effects are not always observed when these HAV strains infect human cell lines. Perez-Rodriguez et al. recently reported that quasispecies genomic selection and molecular breeding using deep-sequencing identified high-fitness improving HAV [48]

Response to your specific comment 2: “Line 106. “Inactivated vaccines are commonly used for HAV infections”. Replaced by “Inactivated vaccines are commonly used to control HAV infections”.”

Thank you for your valuable comments. We agree with you. According to your suggestion, we revised our manuscript as follows.

In line 103,

…Inactivated vaccines are licensed and commonly used to control HAV infections [3]. HAV strain…

Response to your specific comment 3: “Lines 128-129. This sentence would be most adequate as "an alternative strain for a vaccine production" and thus moved to section 2.2. in a context such as “A fast growing strain of HAV with a great potential for antigen production has been isolated by quasispecies genomic selection and molecular breeding [21]”.

Thank you for your valuable comments. We agree with you. According to your suggestion, we revised our manuscript as follows.

In lines 115-118,

As an alternative strain for a vaccine strain, a fast growing strain of HAV with a great potential for HAV antigen production has been isolated by quasispecies genomic selection and molecular breeding [48]. As the production of vaccines require high cost [12], a fast growing HAV strain may be useful for making the production costs of HAV vaccine lower.

Response to your specific comment 4: “Line 132. Section 3. HAV subgenomic replicon, should be replaced by “HAV subgenomic replicon for the study of antiviral dugs”.

Thank you for your valuable comments. We agree with you. According to your suggestion, we revised our manuscript as follows.

In line 132,

  1. HAV subgenomic replicon for the study of antiviral drugs

Response to your specific comment 5:“Line 146. This section should read “Blocking the entry pathway as an antiviral strategy”

”.

Thank you for your valuable comments. We agree with you. According to your suggestion, we revised our manuscript as follows.

In line 142,

  1. Blocking the entry pathway as an antiviral strategy

Response to your specific comment 6: Line 157. Add, the word however, at the beginning of the sentence.

”.

Thank you for your valuable comments. We agree with you. According to your suggestion, we revised our manuscript as follows.

In line 152,

However, there are important discussions on the actual role of HAVcr-1 as a cell surface receptor for HAV [79,80,81]. Costafreda…

Response to your specific comment 7:“Lines 162-163. The sentence “HAVcr-1 is an important attachment factor for eHAV in GL37 cells, although HAVcr-1 is not an 162 essential entry factor for either naked HAV or eHAV [81,82]”, should be better read “In contrast, in other studies the HAVcr-1 is identified as a non-essential entry factor for either naked HAV or eHAV [81,82].”

Thank you for your valuable comments. We agree with you. According to your suggestion, we revised our manuscript as follows.

In lines 164-165,

In contrast, in other studies the HAVcr-1 is identified as a non-essential entry factor for either naked HAV or eHAV [81,82]. Integrin β1 binds…

Response to your specific comment 8:“Lines 168. Add the sentence "Drugs inhibiting the steps of the entry process could be efficient antivirals.”

Thank you for your valuable comments. We agree with you. According to your suggestion, we revised our manuscript as follows.

In lines 172-173,

receptors for HAV are not clearly established yet. Drugs inhibiting the steps of entry process could be efficient antivirals…

Response to your specific comment 9:“Lines167-168. Remove the sentence “However, HAVcr-1, 167 HAV IRES, host cellular and HAV 3C protease, and HAV 3D polymerase are also candidates for anti-168 HAV drug targets (Figure 1) [13, 72-75]”.

Thank you for your valuable comments. We agree with you. According to your suggestion, we deleted this sentence.

Response to your specific comment 10:“Line 171. Section 5. “Evaluation of HAV IRES-mediated translation in human hepatoma cells”, should read “Inhibiting the HAV IRES-mediated translation in human hepatoma cells”.

Thank you for your valuable comments. We agree with you. According to your suggestion, we revised our manuscript as follows.

In line 174,

  1. Inhibiting the HAV IRES-mediated translation in human hepatoma cell lines

Reviewer 2 Report

The review by Kanda et al has been revised partly in line with my comments.  Nevertheless, despite these revisions which have contributed in part to improving the manuscript, there is still a lack of a clear thread connecting the chapters on cell culture models and potential antiviral targets.  A special effort should therefore be made to make this review a single entity. 

I therefore still have some comments which are listed below:

Connecting phrases could be added at the beginning and end of each sub-chapter. 

There are still paragraphs that are out of context and contribute to the lack of clarity of the review.  An example of this is the chapter on HAVcr-1, which, as written in its current form, does not fit into the topic of the review.  The mechanism of endocytosis has been developed in detail, but again the link between paragraph lines 162-169 and cellular models is not clearly established. The last sentence of this paragraph seems to be a general introduction to the concept of anti-viral candidates, and therefore should not be put at the end of this chapter. Finally, there is no clear reference to cellular models to study the entry.  

Some sentences are written awkwardly: for instance, the sentence " HAVcr-1 is an important attachment factor for eHAV in GL37 cells, although HAVcr-1 is not an essential entry factor for either naked HAV or eHAV " should be rewritten, as it says one thing and then the opposite without explaining the reasons why.

The paragraph on vaccines is still not sufficiently detailed and clear.

In this new version of the manuscript, the authors have modified Figure 1 to show the enveloped vs. naked viral particles in the diagram.  However, the way in which the virions are represented is not representative of the situation.  Indeed, the capsid is not represented, which can lead to misinterpretations. 

Author Response

Response to the Reviewer 2: Thank you for your encouraging comments and criticisms.

Response to your comment 1: “Connecting phrases could be added at the beginning and end of each sub-chapter.”

Thank you for your valuable comments. We agree with you. We revised our manuscript as follows.

In lines 97-99,

Cytopathic effects are not always observed when these HAV strains infect human cell lines. Perez-Rodriguez et al. recently reported that quasispecies genomic selection and molecular breeding using deep-sequencing identified high-fitness improving HAV [48].

In line 103,

Inactivated vaccines are licensed and commonly used to control HAV infections [3]. HAV strain…

In lines 115-118,

As an alternative strain for a vaccine strain, a fast growing strain of HAV with a great potential for HAV antigen production has been isolated by quasispecies genomic selection and molecular breeding [48]. As the production of vaccines require high cost [12], a fast growing HAV strain may be useful for making the production costs of HAV vaccine lower.

In line 120,

Despite the use of an effective vaccine, antivirals against HAV would be of great use [12]. The….

In lines 170-173,

…GTPase [79]. The present proposed life cycle model of HAV is shown in Figure 1. Although the mechanism of endocytosis has been developed in detail, their involvement of HAV and cell surface receptors for HAV are not clearly established yet. Drugs inhibiting the steps of entry process could be efficient antivirals.

Response to your comment 2: “There are still paragraphs that are out of context and contribute to the lack of clarity of the review.  An example of this is the chapter on HAVcr-1, which, as written in its current form, does not fit into the topic of the review.  The mechanism of endocytosis has been developed in detail, but again the link between paragraph lines 162-169 and cellular models is not clearly established. The last sentence of this paragraph seems to be a general introduction to the concept of anti-viral candidates, and therefore should not be put at the end of this chapter. Finally, there is no clear reference to cellular models to study the entry. ”

Thank you for your valuable comments. We agree with you. We revised our manuscript as follows.

In lines 164-173,

In contrast, in other studies the HAVcr-1 is identified as a non-essential entry factor for either naked HAV or eHAV [81,82]. Integrin β1 binds elsewhere on the HAV capsid. Trafficking of eHAV to the lysosome is essential for the entry and uncoating of HAV genome and requires the endosomal-sorting complexes required for transport (ESCRT) and programmed cell death 6 interacting protein/apoptosis-linked gene 2 [ALG2]-interacting protein X (PDCD6IP/ALIX), in addition to RAB5A, the member RAS oncogene family (Rab5) and RAB7, member RAS oncogene family (Rab7) GTPase [79]. The present proposed life cycle model of HAV is shown in Figure 1. Although the mechanism of endocytosis has been developed in detail, their involvement of HAV and cell surface receptors for HAV are not clearly established yet. Drugs inhibiting the steps of entry process could be efficient antivirals.

Response to your comment 3: “Some sentences are written awkwardly: for instance, the sentence " HAVcr-1 is an important attachment factor for eHAV in GL37 cells, although HAVcr-1 is not an essential entry factor for either naked HAV or eHAV " should be rewritten, as it says one thing and then the opposite without explaining the reasons why. ”

Thank you for your valuable comments. We agree with you. We revised our manuscript as follows.

In lines 164-165,

…In contrast, in other studies the HAVcr-1 is identified as a non-essential entry factor for either naked HAV or eHAV [81,82]. Integrin β1 …

Response to your comment 4: “The paragraph on vaccines is still not sufficiently detailed and clear. ”

Thank you for your valuable comments. In the previous review of this journal, we deleted and made shorten this section accordingly. We revised our manuscript as follows.

In lines 115-118,

As an alternative strain for a vaccine strain, a fast growing strain of HAV with a great potential for HAV antigen production has been isolated by quasispecies genomic selection and molecular breeding [48]. As the production of vaccines require high cost [12], a fast growing HAV strain may be useful for making the production costs of HAV vaccine lower.

Response to your comment 5: “In this new version of the manuscript, the authors have modified Figure 1 to show the enveloped vs. naked viral particles in the diagram.  However, the way in which the virions are represented is not representative of the situation.  Indeed, the capsid is not represented, which can lead to misinterpretations.”

Thank you for your valuable comments. We agree with you. We revised Figure and its legend accordingly.